

# Longitudinal evaluation of the impact of traditional rainbow trout farming on receiving water quality in Ireland

Alexandre Tahar[1], Alan M. Kennedy[2], Richard D. Fitzgerald[3,†], Eoghan Clifford[2] and Neil Rowan[1]

[1] Bioscience Research Institute, Athlone Institute of Technology, Athlone, Ireland
[2] Civil Engineering, National University of Ireland, Galway, Ireland
[3] Ryan Institute, National University of Ireland, Galway, Ireland
[†] Deceased.

## ABSTRACT

In the context of future aquaculture intensification, a longitudinal ten-year evaluation of the current traditional rainbow trout production in Ireland was performed. Publically available and independent data obtained from local authorities were gathered and analysed. Inlet and outlet concentrations of parameters such as $BOD_5$, ammonium, nitrite, dissolved oxygen and pH for four consecutive flow-through fish farms covering the four seasons over a ten-year period (2005–2015) were analysed. The objectives of the study were (i) to characterize the impact of each fish farm on water quality in function of their respective production and identify any seasonal variability, (ii) to quantify the cumulative impact of the four farms on the river quality and to check if the self-purification capacity of the river was enough to allow the river to reach back its background levels for the analysed parameters, (iii) to build a baseline study for Ireland in order to extrapolate as a dataset for expected climate change and production intensification. For most of the parameter analysed, no significant impact of the fish farming activity on water quality/river quality was observed. These results, the first ones generated in Ireland so far, will have to be completed by a survey on biodiversity and ecotoxicology and compared after production intensification and the likely future introduction of water treatment systems on the different sites.

# INTRODUCTION

With the wild fish catching capacities almost reached and with the overall human population increase on the planet, fish from aquaculture is and will increasingly become a more important food source for human consumption in the near future (e.g., *Donnely, 2011*; *Guilpart et al., 2012*). Aquaculture production increased more than five-fold from 1990 to 2012, while the world capture fisheries increased with only 8% at the same time (*Krause et al., 2015*). Furthermore, it is planned that this increase in production will be sharper in the next decades in Ireland, notably with the objective of increasing food export (including

Corresponding author
Alexandre Tahar, atahar@ait.ie

aquaculture fish products) by 85% by 2025 through the application of the Food Wise 2025, the strategic plan for the development of agri-food sector (*DAFM, 2015*). The future and current intensification of this activity is associated with meeting more stringent environmental regulations to ensure a sustainable and environmentally friendly production. At the European scale, freshwater fish farming is essentially regulated through the Water Framework Directive (WFD) that aims at providing the good chemical and ecological status of the rivers (*Aubin, Tocqueville & Kaushik, 2011*; *European Commission, 2000*; *Guilpart et al., 2012*). Currently, there is a dearth in scientific knowledge on the potential impact and environmental risk of aquaculture farming practices on receiving water quality as it relates to both WFD compliance and related EU Rivers Basin Management Plan 2016–2021.

Various anthropogenic activities can cause the deterioration of river quality including point source pollution such as wastewater treatment plants (WWTPs) with the discharge of partially treated effluents or non-point sources such as agricultural activities (e.g., *Papatryphon et al., 2005*). The traditional freshwater fish farming industry generally operates using flow-through systems (FT) without any water treatment, with the oxygen levels being maintained by relatively high volume water abstraction & flow through the farm. Fish farming activity generates wastes from fish excreta and uneaten feed which if discharged untreated could potentially impact the water quality of the discharge receiving water (*Caramel et al., 2014*; *Garcia et al., 2014*; *Lalonde, Ernst & Garron, 2015*; *Lazzari & Baldisserotto, 2008*; *Sindilariu, Brinker & Reiter, 2009*; *Verdegem, 2013*). The potential impacts of such intensive FT fish farming on receiving river quality can include increased concentrations of five-day biochemical oxygen demand ($BOD_5$), a drop in dissolved oxygen concentrations (DO), an enrichment of total suspended solids (TSS) content in the receiving water, in nutrients such as nitrogen generally characterised by total ammonia nitrogen (TAN), and phosphorus generally characterised by orthophosphate ($PO_4$-P) that could both potentially lead to surface water eutrophication (e.g., *Boaventura et al., 1997*; *Caramel et al., 2014*; *Garcia et al., 2014*; *Lazzari & Baldisserotto, 2008*; *Teodorowicz, 2013*). This impact will depend on the production system employed, the production intensity and on the type of feed but also on the assimilative capacity of the receiving water (*Boaventura et al., 1997*; *Caramel et al., 2014*).

A number of studies have focused on the characterization of freshwater fish farm effluents across the world (e.g., *Boaventura et al., 1997*; *Guilpart et al., 2012*; *Lalonde, Ernst & Garron, 2015*; *Neto, Nocko & Ostrensky, 2015*; *True, Johnson & Chen, 2004*). Most of these studies have inherent drawbacks due to their respective framework; some studies focused only on a short period of time of one year or less (e.g., *Caramel et al., 2014*; *Všetičková et al., 2012*; *Živić et al., 2009*) or relied on individual water samples (e.g., *Caramel et al., 2014*; *Hennessy et al., 1996*; *Lalonde, Ernst & Garron, 2015*; *Noroozrajabi et al., 2013*; *Všetičková et al., 2012*), while the impact of aquaculture should be assessed using longer term evaluation process encompassing production, fish life stages and river characteristics variations (*Aubin, Tocqueville & Kaushik, 2011*; *Hennessy et al., 1996*). Other studies (e.g., *Koçer et al., 2013*; *Sindilariu, Brinker & Reiter, 2009*; *Všetičková et al., 2012*; *Yalcuk, Pakdil & Kantürer, 2014*) have focused on fish farm gates (comparison of farm inlet and outlet water quality) and did not aim at evaluating the impact they might have on receiving water. Some limited studies

(*Aubin, Tocqueville & Kaushik, 2011*; *Boaventura et al., 1997*; *Pulatsu et al., 2004*) analysed the impact of freshwater fish farming on the receiving water quality during a long period of time (i.e., at least two years duration), allowing one to fully understand the impact of specific farms on specific rivers and in specific places. However, data from these studies are difficult to extrapolate and apply into different contexts in light of the parameters and conditions such as species, temperature, aquaculture practices, chemicals used, receiving water features (e.g., flow, hydromorphology) and local environmental conditions that may be very specific to individual farms and locations (*Lalonde, Ernst & Garron, 2015*).

With the increasing demand for fish and fish products worldwide (e.g., *Guilpart et al., 2012*), the freshwater aquaculture industry is facing the challenge of finding the way to produce more without any associated environmental degradation (*Martins et al., 2010*; *Sturrock et al., 2008*; *Teodorowicz, 2013*). In some countries such as Denmark, freshwater aquaculture practices have recently evolved to model trout farms, systems that were able to fulfil both production and environmental objectives (*Jokumsen & Svendsen, 2010*; *Lalonde, Ernst & Garron, 2015*; *Teodorowicz, 2013*). However, some other countries such as Poland (*Teodorowicz, 2013*), France (*Papatryphon et al., 2005*) and Ireland still require a drastic evolution of the aquaculture practices through more advanced systems in order to fulfil those ambitious objectives. To the best of our knowledge, there is no existing data about the impact of freshwater fish farming on downstream river quality in Ireland. There is, therefore, a pressing need to evaluate the impact of aquaculture activity on a longer-term basis using evidence-based data that are publically available, and to ascertain if there is any accumulation of pollution in receiving water. Thus, the main aim of the present study was to evaluate such data in terms of reported aquaculture influent and effluent parameters over a ten year period in order to establish relationship (if any) with receiving water quality in Ireland. In that aim, the impact of four consecutive FT traditional freshwater rainbow trout (*Oncorhynchus mykiss*) farms on the receiving water quality was assessed through publically available historical data analysis (2005–2015). The objectives of the study were (i) to characterize the impact of each fish farm on water quality in function of their respective production and identify any seasonal variability, (ii) to quantify the cumulative impact of the four farms on the river quality and to check if the self-purification capacity of the river was enough to allow the river to reach back its background levels for the analysed parameters, (iii) to build a baseline study for Ireland in order to extrapolate as a dataset for expected climate change and production intensification.

## MATERIALS AND METHODS

### Fish farms description

The studied facilities are all Irish traditional rainbow trout fish farms, all operating in a FT system during the studied period. The schematic representation of fish farms in Fig. 1 shows the very specific configuration of the area with four different farms, all abstracting the water needed and having their discharge into the same river, all in a relative small area (about 4 km between the first and the last farm). This specific configuration allowed the study of the potential cumulative impact the fish farms might have on river quality.

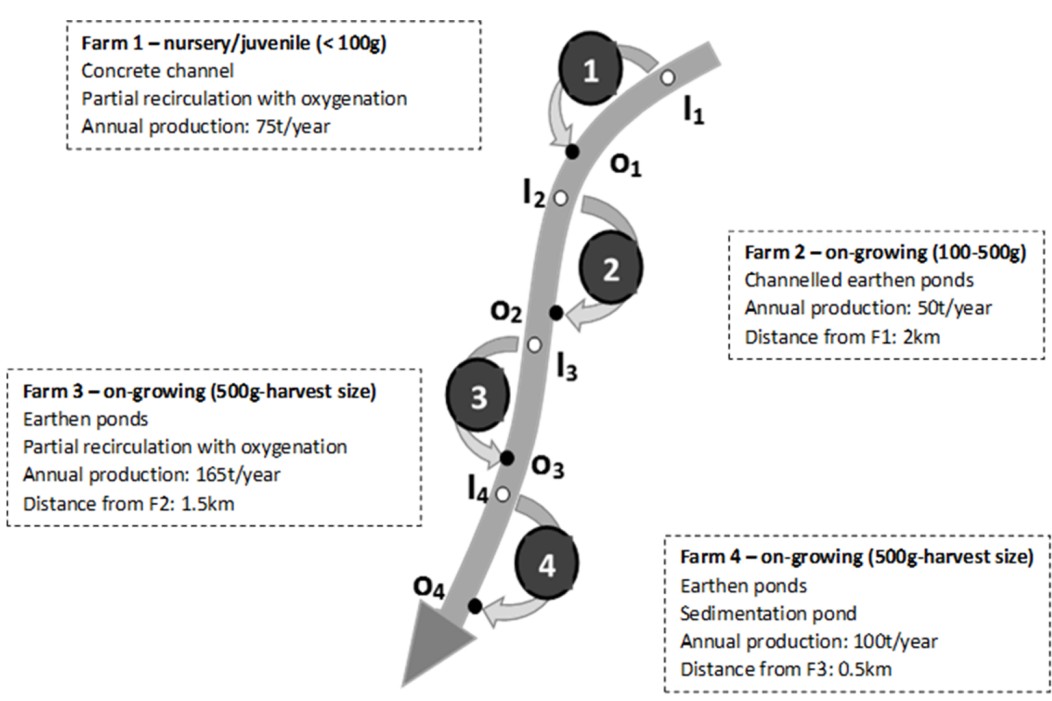

**Figure 1** **Schematic representation of the river, the four studied fish farms and of the associated monitoring points (inlets and outlets).** The main characteristics of each farm are also presented.

Farm 1 (F1) was a hatchery and juvenile (i.e., <100 g) production site. It was operating in FT in a concrete channel separated in different consecutive sections and had a partial recirculation of the water associated to an oxygenation step (no water treatment). As a hatchery site, this farm was the most sensitive to water quality, and which is why it was historically located at the most upstream location where the water quality was not theoretically affected by the fish farm activity yet. Farm 2 (F2) was about 2 km downstream from F1 and was a medium size (i.e., 100–500 g) on-growing site that was operating in FT in an earthen pond. Farm 3 (F3) was about 1.5 km downstream from F2 and was an on-growing site (i.e., 500 g to harvest size) that was operating in FT in earthen ponds with a partial recirculation of the water associated to oxygenation (no water treatment). Farm 4 (F4) was about 500 m downstream from F3 and was an on-growing site (i.e., 500 g to harvest size) that was operating in FT in an earthen pond associated with a sedimentation pond aiming at removing settleable solids from the discharge water.

On average, the four farms produced 75, 50, 165 and 100 tons a year respectively during the investigated period (2005–2015).

Considering that there was no or limited form of water treatment on these farms (Fig. 1), the objective was to evaluate the impact of the whole pool of farms, and of successive individual farms, on the water quality in order to check if there was any accumulation of pollution (e.g., TSS, $BOD_5$, TAN, $PO_4$-P) within their receiving water. According to Ireland's EPA (http://www.epa.ie/hydronet/#Water%20Levels), the river 95% percentile flow was calculated as being 1.1 $m^3$/s (average over 30 years of 5.5 $m^3$/s). It is noteworthy
that there was no agricultural activity or WWTPs within the area (i.e., upstream and in between the different farms) and that therefore, in the present study, any significant modification of the river quality downstream from the farms will be attributed to the fish farming activity and will represent its impact on the river quality.

The farms water intake flow was measured at regular intervals by the fish farmer and were about 0.4 $m^3$/s on average on each farm (range 0.1–0.6 $m^3$/s depending on the farm and the flow conditions in the river).

## The discharge licence

Each Irish County Council (Local Authority) require two licences to operate—an aquaculture licence issued by the Department of Agriculture, Food & the Marine and a Trade Effluent Discharge Licence issued by the relevant Local Authority.

For these four farms the discharge licences states both (i) a regulation on the maximum water abstraction rate that the farm cannot exceed depending on the river flow, and (ii) a maximum differential concentration (or value) between fish farms influent and effluent waters for a range of parameters (i.e., 1 $mgO_2$/L for $BOD_5$, 10 mg/L for TSS, 5 NTU for turbidity, 0.4 mgN/L for TAN, 0.002 mgN/L for nitrite ($NO_2$-N) and 0.2 mgP/L for $PO_4$-P). For three other parameters, the regulation defines absolute limit values for the farm effluent (i.e., 60% saturation for DO, range 6–9 for pH and ambient temperature). The discharge licence specified that each fish farm had to be sampled by the regulatory agency for both inlet and outlet water at a frequency of four times a year (i.e., generally one sampling per season).

## The parameters monitored

A full ten-year record of historical data (2005–2015) generated by an independent and accredited water analysis lab was gathered for this study. The present study focused on the parameters that were monitored and analysed during the studied period. Hence regulated parameters (presented in the previous section) and total oxidised nitrogen (TON), representing the sum of $NO_2$-N and nitrate ($NO_3$-N), were considered for the present study.

Surprisingly, according to the publicly available documents that were consulted for this study, ammonium ($NH_4$-N) was monitored instead of TAN (regulated parameter) that represents the sum of $NH_4$-N and ammonia ($NH_3$-N), the latter being much more toxic for fish (*Tomasso, 1994*). The predominance of one ammonia form or another depends on pH and temperature with $NH_3$-N predominating at high temperature and pH values (*Emerson et al., 1975*). Therefore, for the present study, pH and temperature values were used to enable an estimation of TAN concentrations from $NH_4$-N measurements.

Validated standard methods (*APHA, 2012*; *APHA, 2005*) were employed by the accredited labs for the analysis of each monitored parameter.

## Sampling

All samples were performed as spot samples and no composite samples were taken. According to the Local Authority responsible for the water monitoring, all the inlet sampling locations ($i_{1-4}$, see Fig. 1) were located in the river itself immediately upstream

from the inlet channel of each farm (taken every time at the same section at the centre of the river). The outlet sampling spots ($o_{1-4}$, see Fig. 1) were directly located in the outlet channels of each farm (farm effluents) and therefore do not include any dilution by the downstream river.

The discharge licence did not specify a sampling method and thus it is likely that spot samples were employed as they were considered the most efficient and cheap sampling method. Samples (i.e., 1L in PET bottles) were stabilized in acidic conditions (to reach a pH value below 2 in order to avoid any transformation of the nitrogenous compounds) and brought to the labs where they were kept refrigerated before analysis. Dissolved oxygen saturation levels, pH and temperature values were obtained *in-situ* before sample stabilization by a multi-parameter sensor (YSI 51B oxygenmeter and WTW pH 330 pH meter). Separate samples were taken for the analysis of $BOD_5$ parameter (not acidified).

## Approach employed for the quantification of the fish farms impacts

The following range of assessments were considered when evaluating the impacts of the farms on the receiving river water—individual farm impact on water quality, and cumulative impact of the four fish farms on river water quality. Additionally, in order to check a potential higher impact during summer condition (as observed in *Lalonde, Ernst & Garron, 2015*), the seasonal variability of the impacts was also studied by considering the evolution of the effluents, quality and the impacts for the different seasons during the year. For this purpose, data obtained from a monitoring in December, January and February were classified as ''winter data''; the ones from the months of March, April and May as ''spring data''; the ones from the months of June, July and August as ''summer data'' and the ones from the months of September, October and November as ''autumn data''.

As discussed above, the four fish farms were abstracting the water from and discharging into the same river (Fig. 1). Therefore, the fish farms inlet analysis results (i.e., $I_1$, $I_2$, $I_3$ and $I_4$) were employed to study the evolution of the water quality along the river across the 4 farms. We noticed that this method was used to study the impact of the three first farms (i.e., farms 1–3). The evaluation of the cumulative impact of the three first farms (F1, F2 and F3) on the river quality was achieved by the comparison of the results obtained in $i_1$ and $i_4$ sampling locations. This allowed us to take account of the dilution of the effluent of each farm by the river and therefore to take account of the pollution load emitted by the farms. The impact of F4 was not possible to assess this way because river quality downstream from this farm was not monitored (i.e., the only location of these farm effluent monitoring points that does not take account of the dilution by the river, see Fig. 1).

Unfortunately, those results were not correlated with the fish parameters because we could not have access to accurate records of these biological data but only of average yearly production. Therefore, this study only focused on water quality parameters.

## Data treatment

The following equation was used to calculate the differential concentrations ($D_i$) for each monitored farm (i) and parameter introduced above.

$$D_i = O_i - I_i.$$

With $O_i$ and $I_i$ the outlet and inlet concentrations at the farm $i$ for a given parameter on a given sampling event.

Only concentration values were considered and treated although load values (taking farm outlet flow values in consideration) would have been a better way to characterize the impact of the different farms. However, flow values were not measured continuously on the different sites and this did not allow for an accurate calculation of the loads for the different parameters at different times. Also, river concentrations were employed to characterize the impact of the different farms on river quality; those river concentrations are taking river dilution into account and were therefore considered as a good approach to consider the load of each of parameters considered.

Statistical tests were performed using MATLAB software (MathWorks, Natick, MA, USA) in order to assess if datasets means were significantly different (e.g., concentrations measured upstream and downstream from each farm) and if the fish farms had an impact on the water quality. Box plot representations of the data were chosen in order to show the dispersion of the data and to avoid any misinterpretation due to the occurrence of extreme values. Two samples Student's $t$-tests were employed to compare datasets after checking the normal distribution of each dataset (validity domain of this statistical test). A confidence interval of 95% was systematically employed.

Many results from the datasets obtained from the regulation body were below the limit of quantification (LOQ) associated to the analytical method employed. In this case, for the calculations performed in the present study and because the sampling and analysis were not performed by our team, the data was considered as equal to the LOQ (substitution approach). The real value of the data could be lower than LOQ but this "conservative" approach of "below LOQ = LOQ" was chosen here, although other approach could have been used such as "below LOQ = LOQ/2" or "below LOQ = LOQ/$\sqrt{2}$" (*Hornund & Reed, 1990*; *US Environmental Protection Agency, 2003*), to avoid any misinterpretations.

## RESULTS AND DISCUSSION

### General overview of the results—outcome of inlet and outlet values for each parameters

More than 1,000 inlet/outlet couples of data were gathered overall for all the parameters and the four fish farms investigated. The full dataset as well as the representation of the data for each parameter and each farm and the monthly distribution of the data are presented in supplementary materials sections (SP1, Tables S1 and S2).

Figure 2 gives an overview of the gathered data for all farms and presents a global comparison of inlet and outlet water quality observed globally and for all the parameters monitored.

Globally, by comparing inlet and outlet values for each parameter, Fig. 2 shows the mean increase or decrease of the monitored parameters values caused by the four farms without distinction. No significant differences between farms inlets and outlets were observed for TON, temperature, BOD$_5$, turbidity, TSS and NO$_2$-N ($P$ value > 0.05) demonstrating that the investigated fish farms might have no impact on downstream river quality for these

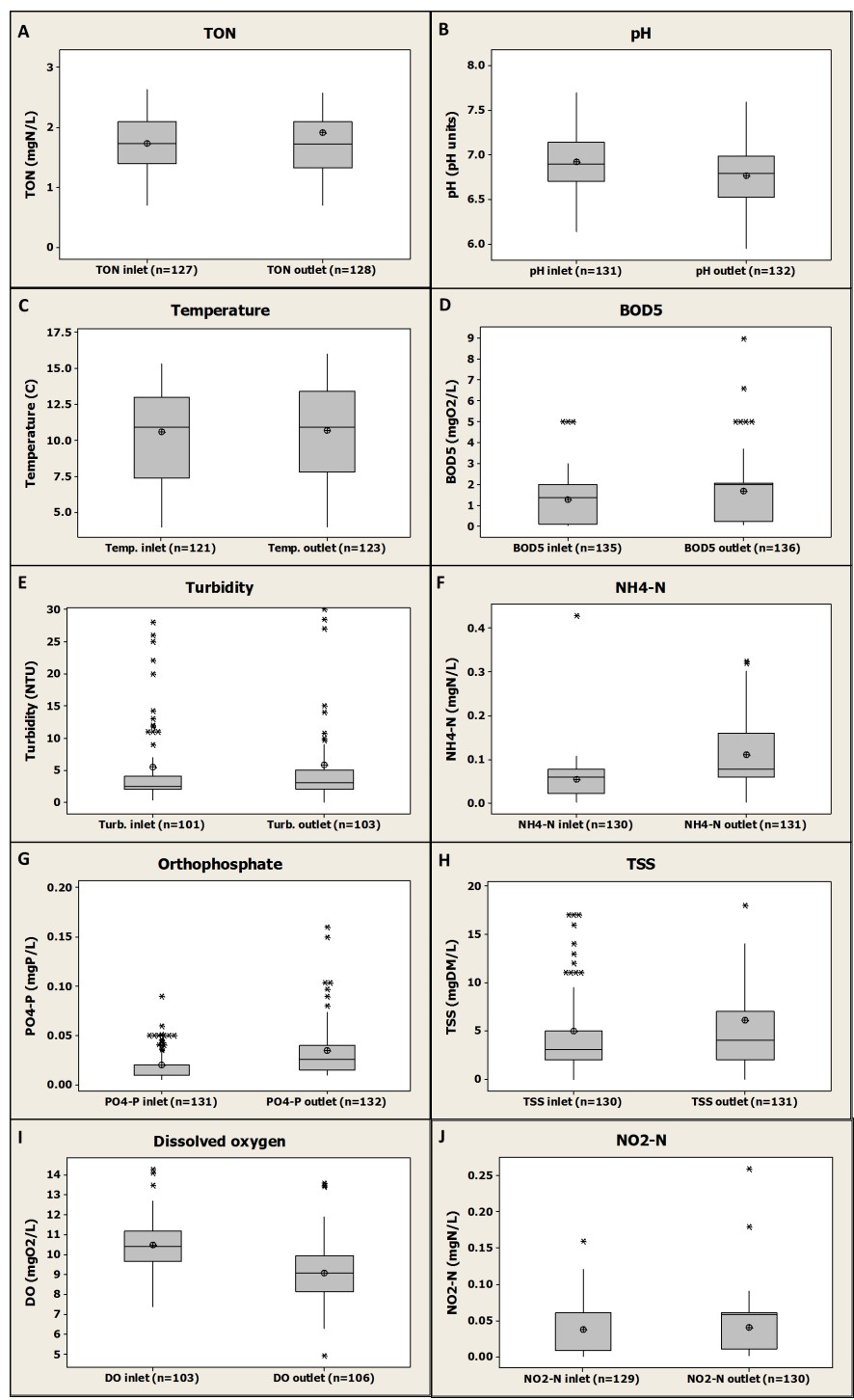

**Figure 2** Box-plot global representation of inlet and outlet values compiled and gathered for the four fish farms and for all the monitored parameters (2005–2015).

parameters. However, a significant increase from fish farms inlets to outlets was observed for $NH_4$-N ($P$ value $< 0.001$), $PO_4$-P ($P$ value $< 0.001$) and $BOD_5$ ($P$ value $< 0.01$) demonstrating that the investigated fish farms might have an impact on the downstream river quality through the release of those compounds. However, the increase in $NH_4$-N was only about 0.08 mgN/L on average ($\pm 0.08$ mgN/L, $n = 130$, see Fig. 2F), which can be considered as a relatively low value compared to values found in other studies such as a 1.46 mgN/L increase (*Boaventura et al., 1997*). The same observation can be done for $PO_4$-P with an average increase of about 0.015 mgP/L ($\pm 0.03$ mgP/L, $n = 130$, see Fig. 2G) found in the present study and a range of 0.06–0.58 mgP/L increase found in another study where three different rainbow trout farms were investigated during one year (*Boaventura et al., 1997*). Some other studies observed some increase from rainbow trout farms inlets to outlets for $NH_4$-N (*Caramel et al., 2014*; *Guilpart et al., 2012*; *Kırkağaç, Pulatsu & Topcu, 2009*; *Lalonde, Ernst & Garron, 2015*), $BOD_5$ (*Teodorowicz, 2013*), total nitrogen (*Caramel et al., 2014*; *Lalonde, Ernst & Garron, 2015*), total phosphorus (*Caramel et al., 2014*; *Kırkağaç, Pulatsu & Topcu, 2009*; *Lalonde, Ernst & Garron, 2015*) and $PO_4$-P (*Caramel et al., 2014*; *Guilpart et al., 2012*). Furthermore, a significant decrease was observed in the present study at fish farms outlet compared to inlets for both DO ($P$ value $< 0.001$) and pH ($P$ value $= 0.001$) showing, as expected, (i) a global oxygen consumption due to fish metabolism (*Boyd & Tucker, 1998*), and (ii) a production of carbon dioxide by fish with the consequence of lower pH value at the outlets (*Boyd & Tucker, 1998*). An average decrease in DO of about 1.42 $mgO_2$/L ($\pm 1.28$ $mgO_2$/L, $n = 101$, see Fig. 2I) from farms inlets to farms outlets was observed, that is in agreement with previously published results such as a study where a DO decrease of 0.7–2.4 $mgO_2$/L depending on the fish farm investigated was found (*Boaventura et al., 1997*). A deeper focus on these data, with an emphasis on the inlet and outlet of each farm, will be presented in the following sections of this study as to whether or not they confirm these global trends. An evaluation of potentially significant impacts of those individual fish farms on the receiving water quality for the monitored parameters will be provided.

### Impact on water quality—individual farms level

The objective of this section are (i) to quantify the impact of each farm on water quality and (ii) to address their compliance to their discharge licences criteria.

#### Parameters regulated in term of differential concentrations

Here we present an outcome of the differential concentrations for five regulated parameters (i.e., $BOD_5$, $NO_2$-N, $PO_4$-P, TSS and turbidity) and for ammonium ($NH_4$-N) for each farm (Fig. 3).

A first general observation of the results is that for most of the regulated parameters (i.e., turbidity, TSS, $PO_4$-P, $NH_4$-N, $BOD_5$) neither the average nor median values are higher than their associated differential limit value. The different farms were therefore globally in compliance with their discharge licence for those parameters. An exception is $NO_2$-N, with an average differential concentration higher than the limit value (i.e., $0.006 \pm 0.025$ mgN/L, $n = 40$ vs 0.002 mgN/L, see Fig. 3C) for F4. However, the median value was equal

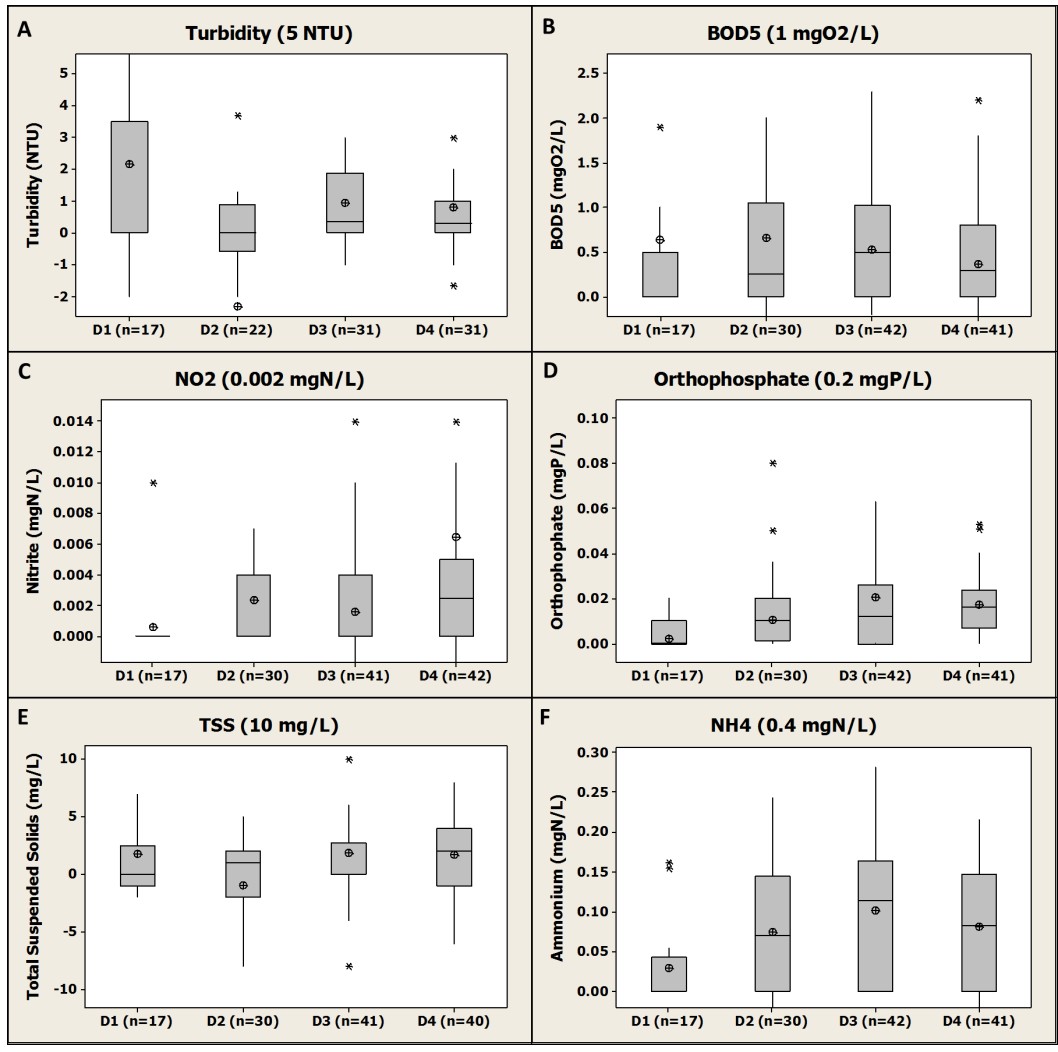

**Figure 3** Box-plot representation of the differential concentrations ($D_i$) between inlet and outlet of each of the four fish farms and for each regulated parameter (2005–2015). Numbers 1, 2, 3 and 4 represent the consecutive fish farms in the order of the river flow direction. In brackets, the discharge licence limit value for each regulated parameter as mentioned in the discharge licence. Median, average, Q1 and Q3 are presented. For $NH_4$-N, the limit value associated is for total ammonia (TAN).

to the limit value (i.e., 0.002 mgN/L) underpinning the influence of extreme values on the average calculation. For this parameter, a large number of data were <LOQ and as stated above were considered as equal to the LOQ; for F1 a very large proportion of the data (i.e., about 95%) were <LOQ at both inlet and outlet giving resulting differential concentrations of 0 mgN/L according to the equation presented in the materials and method section (it is noticed that the same result would have been obtained with a different choice of values for analysis below the LOQ). Surprisingly, the LOQ for $NO_2$-N was generally 0.06 mgN/L which is higher than the differential concentration limit of 0.002 mgN/L. This shows the limit of the approach set by the Local Authority to set some very low differential limits

that cannot actually be addressed by the chemical analysis procedure typically applied in accredited chemical analysis labs.

For $BOD_5$, a global compliance with the discharge licence was observed with both average and median values below the differential limit value (i.e., 1 $mgO_2$/L, see Fig. 3B) for the four farms. However, for F2, F3 and F4 a substantial number of individual differentials values was above the limit value (i.e., about a quarter of the values for the three farms) demonstrating that even if a general compliance was observed the compliance was not achieved all the time. The low differential limit value for this parameter (i.e., 1 $mgO_2$/L) compared to other fish farms in the country (i.e., generally 2 mg/L) might explain this relative high number of "non-compliance" data. Furthermore, according to the statistical analysis performed, no significant difference were observed between the different farms for $BOD_5$ differential values (i.e., $P$ values $> 0.05$).

For TAN (i.e., $NH_4$-N + $NH_3$-N), the compliance cannot be directly assessed by the available $NH_4$-N monitoring; however, considering that pH values were always below 8 (i.e., range 6.5–7.5 pH units, see Supplementary Material SP1) and the temperature range of 5–15 °C during all the monitoring period (see Supplementary Material SP1), $NH_4$-N was highly predominating over $NH_3$-N and therefore TAN concentrations were similar to $NH_4$-N (about 0.1 mgN/L depending on the farm, see Fig. 3F). Therefore, the conclusion is that all farms were in compliance with the differential concentration limit for TAN of 0.4 mgN/L. Furthermore, according to the statistical analysis performed, some significant differences were observed between the different farms in terms of $NH_4$-N differential values; mean $NH_4$-N differential value for F1 was revealed to be significantly lower than for F2 ($P$ value $> 0.05$), F3 ($P$ value $< 0.001$) and F4 ($P$ value $< 0.01$). No significant difference was observed between F2, F3 and F4 for this parameter.

Farm 2 and F4 showed a substantial proportion of negative differential values for turbidity and TSS (i.e., nearly half of the total number of data for these farms, see Fig. 3E) demonstrating that those farms were polishing the water regarding solids. The earthen pond configuration and low water velocity for F2 and the presence of a sedimentation pond in F4 could explain this observation (Fig. 1). However, the presence of such solid removal processes did not enhance the removal of other compounds such as $BOD_5$ and $PO_4$-P that could have been reduced by the presence of a sedimentation pond (*Teodorowicz, 2013*). The statistical analysis of the data showed that the TSS differential mean value was significantly lower for F2 than for F3 ($P$ value $< 0.01$). No other significant difference between farms was observed for both TSS and turbidity.

Farm 3 was the farm with the highest average yearly production of 165 tons (Fig. 1) and was also the one associated to the highest average differential values for $PO_4$-P and $NH_4$-N (i.e., 0.02 $\pm$ 0.042 mgP/L, $n = 42$ and 0.1 $\pm$ 0.08 mgN/L, $n = 42$, respectively, see Figs. 3D and 3F), even if the difference with the other farms were not significant according to the statistical analysis. In the present study, a higher fish production led to a higher release of phosphorus and nitrogenous wastes as was observed in other studies such as *Guilpart et al. (2012)* where a proportionality relationship was observed between nutrient release and farm production. For the same set of parameters, F1 was revealed to be less impacting than the other farms with on average lower differential values than any other farm. This might

be due to a relative low yearly production in this farm (i.e., 75 tons, see Fig. 1) and the fact that this was a hatchery and juvenile production farm as it was observed in another study (*Guilpart et al., 2012*; *Teodorowicz, 2013*). The statistical analysis revealed that mean $PO_4$-P differential value for F1 was significantly lower than for F3 ($P$ value $= 0.01$) and for F4 ($P$ value $< 0.001$). No other significant difference between the different farms was observed for $PO_4$-P.

Globally, a general compliance with the discharge licence for all the regulated parameters was observed for the four farms and therefore the studied individual fish farms were not substantially impacting water quality. However, it was also observed that the F3 and F4 had on average higher differential values than the other farms. This fact could be due to higher yearly production for F3 and F4 compared to the other farms (Fig. 1) (*Boaventura et al., 1997*; *Guilpart et al., 2012*; *Teodorowicz, 2013*).

### *Parameters regulated in term of absolute values*

Three parameters were regulated in term of absolute effluent values (i.e., DO, pH, temperature). The limit value for DO was 60% saturation in water and it appeared in the present study that no value was found to be below this threshold (average value for all farm outlets of 80% saturation, see Supplementary Materials SP1). The accepted range for pH was 6–9 pH units and no value was found to be out of this range, with an actual range of 6.5–7.5 pH units for all farms (see Supplementary Materials SP1). For temperature, the discharge licence stated that the effluent had to be "ambient" and no significant differences were observed between farms inlets and outlets (see Supplementary Materials SP1) meaning that the outlet water was actually at ambient temperature. Therefore, the four different farms were in compliance with the discharge licence for these three parameters and the farms were not substantially impacting water quality for these parameters.

The statistical analysis revealed that for temperature, the mean differential value for F3 was significantly higher than for F4 ($P$ value $< 0.05$). This could be due to the different configuration of those farms; F3 being a multi-pond based farm with an assumed (not measured) associated relatively high hydraulic retention time (HRT), allowing a higher water heating potential than F4 which was assumed to be associated to a lower HRT due to its different configuration (i.e., only two ponds) and therefore a lower heating potential than F3. The statistical analysis for the other parameters (i.e., pH and DO) mean differential values did not reveal any significant differences between the four farms.

## Cumulative impact of the fish farms on the river quality

The evolution of the quality of the inlet water through the four consecutive fish farms is presented in Fig. 4.

### *Upstream water quality*

Background river quality (i.e., $I_1$, see Fig. 4) was on average very suitable for rainbow trout production with temperature range of 5–15 °C (see Supplementary Materials SP1), high level of DO (i.e., about $10.3 \pm 1.29$ $mgO_2$/L, $n = 17$ on average, see Fig. 4G), low levels of $BOD_5$ (i.e., about $2.06 \pm 0.95$ $mgO_2$/L, $n = 17$ on average, see Fig. 4B), $NH_4$-N (i.e., $0.08 \pm 0.004$ mgN/L, $n = 17$ on average, see Fig. 4D), $NO_2$-N (i.e., $0.06 \pm 0.012$ mgN/L, $n = 17$

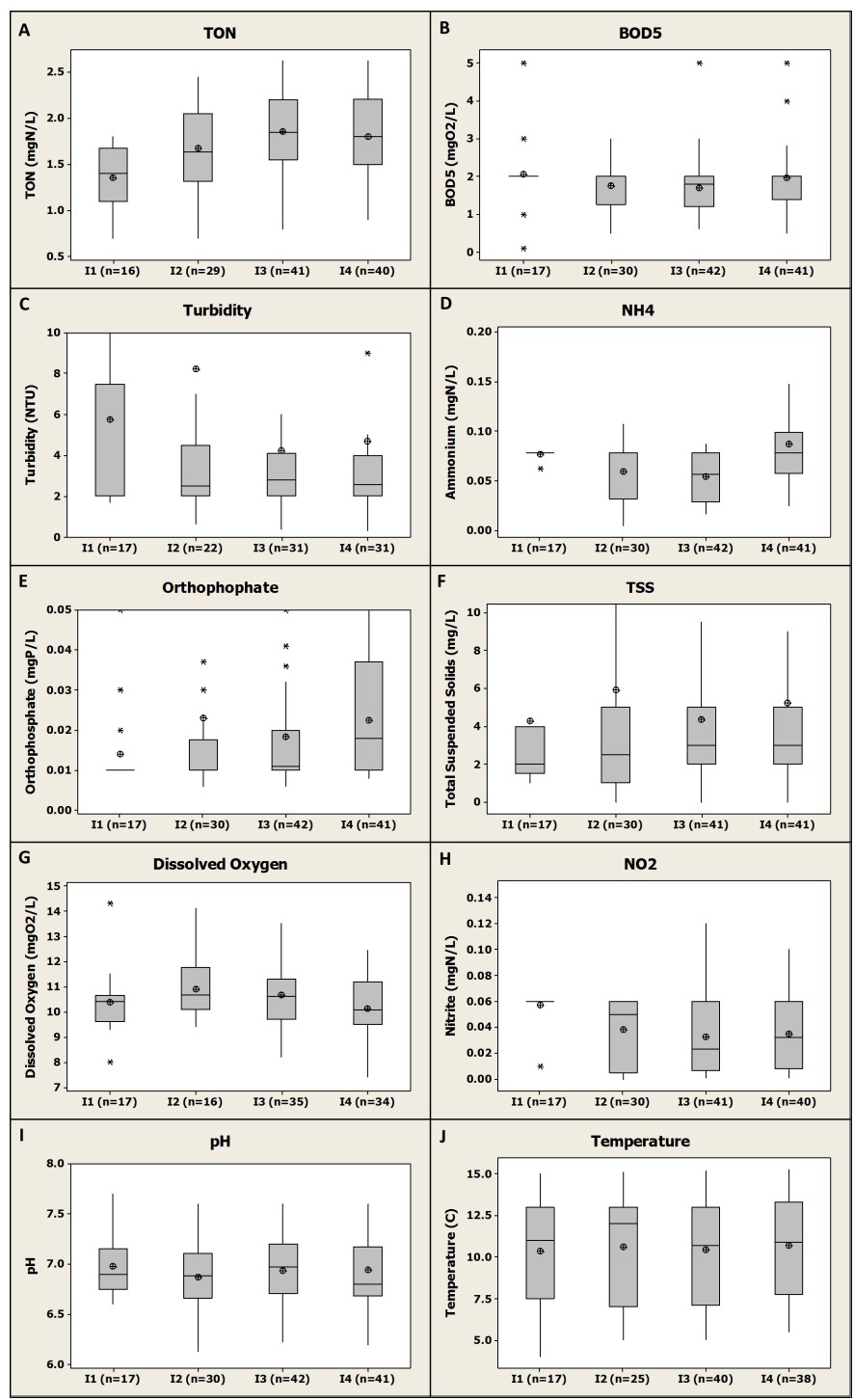

**Figure 4** **Box-plot representation of the water quality evolution through the four different fish farms inlets showing the cumulative impact of the fish farms on river quality for the monitored parameters.** Numbers 1, 2, 3 and 4 represent the consecutive farms in the order of the river flow. Median, average, Q1 and Q3 are presented.

on average, see Fig. 4H) and TON (i.e., $1.36 \pm 0.32$ mgN/L, $n = 16$ on average, see Fig. 4A). Overall, these levels are all suitable for fish growth and all below the proven chronic toxicity levels for salmonids (*EEC, 1978*). Furthermore, these values confirmed that there was no potentially polluting activity (e.g., agriculture, WWTP) upstream from the fish farms that would have degraded the river quality for the parameters monitored.

### BOD$_5$/dissolved oxygen (DO) patterns through the river flow

Overall there was no negative impact observed on river water quality due to the farms in terms of BOD$_5$ and DO concentrations which remained at about 2 mgO$_2$/L and 10 mgO$_2$/L respectively across the four different farms inlets and therefore along the river.

For DO, these results demonstrated that even if a significant decrease was observed at farm scale (as mentioned before), the river had the potential to reoxygenate between the different farms to reach its background level. In a similar study dealing with the cumulative impact of five different rainbow trout farms in Turkey, *Pulatsu et al. (2004)* observed an impact of fish farming that significantly decreased the DO levels in the river at 100 m downstream from the last farm. This relative short distance compared to distances between the different farms in the present study might explain why an impact was observed in this study.

For BOD$_5$, no significant difference in the inlet mean concentrations for the four farms was observed in the present study according to the statistical test applied ($P$ value $> 0.05$) (Fig. 4B). This demonstrated that even if some BOD$_5$ differential concentrations were higher than the limit values for F2, F3 and F4, this had globally no impact on the river quality. A possible explanation for this observed trend could be a combination of (i) a relative high river flow compared to average fish farms water uptake flow providing a high dilution capacity to the river, (ii) a relative low differential concentrations for BOD$_5$ associated to river self-cleaning potential for this compound as it was observed in a study dealing with nine different rainbow trout and carp farms in Poland (*Teodorowicz, 2013*). On the contrary, *Pulatsu et al. (2004)* observed a significant impact of trout farming on the BOD$_5$ concentration in the river. However, in this study the downstream monitoring station was only at 100 m from the last farm investigated that might not be enough distance for the river to clean itself or to reoxygenate as it was observed in other studies (e.g., *Boaventura et al., 1997*; *Teodorowicz, 2013*). Depending on river/farm flow and river characteristics, a distance of 2–3 km downstream the fish farm is considered as being necessary to allow a self-purification of the river regarding BOD$_5$ (*Boaventura et al., 1997*).

### Ammonium pattern through the river flow

Slight variations were observed for NH$_4$-N concentrations across the four fish farms inlets and therefore along the river. However, upstream and downstream farms inlets (i.e., F1 and F4 respectively) were associated to similar NH$_4$-N mean concentrations of 0.08 mg/L (no significant differences between those two farms, $P$ value $> 0.05$) (Fig. 4D). Thus, as for BOD$_5$ parameter, we can assume that the distance between the different farms in the present study was long enough to allow the river to purify itself regarding NH$_4$-N and to get back to its initial background concentration. Therefore, there was not any impact of the fish farms on the river quality in term of NH$_4$-N concentrations. However a significant difference

was observed between F3 and F4 for this parameters ($P$ value $< 0.01$) with a higher value at F4 inlet meaning that F3 might have had an impact on river quality for $NH_4$-N. The relative high average yearly production in F3 (i.e., 165 tons, see Fig. 1) associated to the relative short distance between those two farms (i.e., 500 m, see Fig. 1) might explain this observation due to a limited self-cleaning potential by the river within this relative short distance (*Boaventura et al., 1997*; *Lalonde, Ernst & Garron, 2015*). Depending on farm production and dry weather flow for the receiving water, *Boaventura et al. (1997)* estimated at 3–12 km the distance necessary for the river to get back to its initial organic chemical compound (including $NH_4$-N) concentration downstream from a given trout farm. With such low differential concentrations in F4 ($0.08 \pm 0.08$ mgN/L, $n = 41$, on average, see Fig. 3F) and the self-cleaning potential of the river for this compound (*Boaventura et al., 1997*; *Lalonde, Ernst & Garron, 2015*), it appears very unlikely that this fish farm would have any impact on the downstream river quality in term of $NH_4$-N concentration.

### Nitrite pattern through the river flow

A decrease in $NO_2$-N concentrations was observed across the fish farm inlets and therefore along the river (Fig. 4H). Median $NO_2$-N concentration was revealed to be 0.06 and 0.03 mgN/L at the upstream (i.e., F1 inlet) and downstream (i.e., F4 inlet) monitoring locations respectively. Therefore, considering this observation, the fish farm activity might have had a positive impact on the river quality regarding the $NO_2$-N concentration. To our knowledge, this potential "purification" of a river by trout farms regarding $NO_2$-N was never observed in fish farms where no denitrification step is applied, as was the case for the investigated fish farms in the present study where no water treatment processes were applied.

However, this result might be an artefact because as observed before most of the results obtained for F1 inlet location were below the LOQ of generally 0.06 mgN/L. As previously mentioned, in these cases, a value equal to the LOQ was considered for the present study but the real values could have been lower. The analytical LOQ was generally lower for F4 than for F1; this could explain why this "apparent dilution" of the river by the farms effluent was obtained for this parameter. The relative small dataset for F1 ($n = 17$) compared to the other farms ($n = 30$–41) might also explain the differences observed between upstream and downstream locations (i.e., F1 and F4 respectively). Other studies demonstrated the potential impact of trout farming on $NO_2$-N levels in river. *Pulatsu et al. (2004)* observed a significant increase from 0.019 to 0.581 mgN/L from upstream to 100 m downstream from five different trout farms. This could confirm the occurrence of an artefact with the results of the present study.

### Total oxidized nitrogen pattern through the river flow

A slight increase was observed across the fish farms inlets and therefore along the river for TON with median concentrations of 1.4 and 1.8 mgN/L for the upstream (i.e., F1) and downstream (i.e., F4) locations respectively (Fig. 4A). Therefore, fish farming might have a slight impact on the river quality considering this parameter and might be responsible for a release of $NO_3$-N (not directly monitored). However, according to statistical analysis, no significant difference was observed between F1 and F4 inlets for this parameter average

values ($P$ value $> 0.05$). This result is in agreement with the literature on this topic; in one study, $NO_3$-N mean concentrations were observed to increase from 0.13 mgN/L to 0.43 mgN/L from upstream to downstream of freshwater salmonids farms in Canada but no significant difference was observed (*Lalonde, Ernst & Garron, 2015*). In another study, a significant impact of trout farming on river quality for both $NO_3$-N and $NO_2$-N was demonstrated (*Pulatsu et al., 2004*). Another study dealing with the potential impact of eight different trout farms in France demonstrated that there was no trend for $NO_3$-N and that fish farms could be either responsible of an increase or a decrease of $NO_3$-N concentrations in the receiving water (i.e., 100, 1,000 m downstream from the farms) (*Guilpart et al., 2012*).

### Orthophosphate pattern through the river flow

Stable $PO_4$-P concentrations were observed from F1 inlet to F3 inlet. Then, an increase was observed on $PO_4$-P concentration from 0.01 mgP/L to almost 0.02 mgP/L between F3 and F4 inlets respectively (i.e., median values, see Fig. 4E). Therefore, and considering that there was not any other potentially polluting activity in the area (i.e., WWTP, agriculture), F3 had an impact on the river quality in term of $PO_4$-P concentration. This was confirmed by the statistical analysis revealing that mean $PO_4$-P inlet concentrations for F3 and F4 were significantly different ($P$ value $< 0.001$) and that $PO_4$-P concentrations were significantly higher at the F4 inlet than at F3 inlet. As for $NH_4$-N this trend might be due to both a relative high production in F3 and short distance between F3 and F4. Considering similar $PO_4$-P differential concentrations for F3 and F4 of about 0.02 mgP/L (see Fig. 3D) and a similar water uptake flow both farms, it is highly possible that an impact of F4 would have been observed if the river was monitored directly downstream from this fish farm. However, considering the self-cleaning potential of the river, the river might have got back to its $PO_4$-P background concentration of about 0.01 mgP/L (see Fig. 4E) a few kilometres downstream from F4 (*Boaventura et al., 1997*). In another study, an impact of trout farming on total phosphorus (TP) on the river quality was observed with an increase from 0.069 mgP/L upstream to 0.117 mgP/L downstream from the fish farms (*Pulatsu et al., 2004*). It is assumed that this increase in TP is likely to be due to an increase in the reactive form, $PO_4$-P, which was the monitored parameter in the present study.

### Total suspended solids/turbidity patterns through the river flow

A slight TSS (and turbidity) increase were observed all along the river from 2 mg/L (2 NTU) to 3 mg/L (2.5 NTU) for upstream (i.e., F1 inlet) and downstream (i.e., F4 inlet) locations respectively (Figs. 4F and 4C). This demonstrated that the four fish farms had a very limited impact on the river quality in term of TSS (and turbidity). This limited impact was confirmed by the statistical analysis that demonstrated that there was no significant differences between TSS and turbidity average values between F1 and F4 inlet values ($P$ value $> 0.05$). Considering relative low TSS and turbidity differential values observed for F4 (see Fig. 3E), it is unlikely that this fish farm would impact the downstream river quality for these parameters. This result is in agreement with the literature (*Lalonde, Ernst & Garron, 2015*; *Pulatsu et al., 2004*) where a non-significant impact of trout farms on receiving water quality was observed for TSS.

### Temperature/pH patterns through the river flow

No significant impact of fish farms on both river temperature and pH was observed in this study. This result is in agreement with *Pulatsu et al. (2004)*, who did not observe any impact of trout farms on the receiving water quality for these parameters.

## Seasonal variability (influent/effluent quality, impact)

Differential values were gathered by season with no distinction between the different farms. The results are presented in Fig. 5.

The results were compared statistically season by season in order to show any significant difference from one season to another for each parameter. Overall, except for $NH_4$-N (Fig. 5C), no significant difference was observed between the average results obtained for each season for all the parameters monitored. Even if differential values were on average higher in summer for TON, $NH_4$-N, $PO_4$-P and $NO_2$-N, the statistical analysis revealed that no significant differences were observed with mean values calculated for the other seasons. The analysis of the results obtained for $NH_4$-N revealed that mean values obtained for the spring season were significantly lower ($P$ value < 0.01) than the results obtained for any other season. No significant difference was observed when the other seasons were compared. These results demonstrate that the season change had an overall limited impact on the polluting potential of the fish farms investigated and on the parameters monitored and that, despite apparent higher differential values for nutrient during summer, fish farming was not significantly more impacting during this season than during any other seasons in terms of discharge. This observation is in agreement with another study where a negligible impact of season on 15 trout farms effluent quality was observed for all the parameters monitored except for TP (*Lalonde, Ernst & Garron, 2015*). However, other studies demonstrated that the impact of trout farms on receiving water quality was higher in summer conditions when stocks are high, and both river flow and DO levels are low (*Boyd & Tucker, 1998*). In the future, climate change might increase the gap between winter and summer conditions with potential higher temperature/lower flows in rivers during the summer; therefore, the impact on aquaculture and on effluent water quality might be more and more significant.

## CONCLUSION

A ten-year longitudinal survey (2005–2015) of the impact of four consecutive FT trout farms was performed for the first time in Ireland based only on publically available and independently generated data analysis. First, it was demonstrated that publically available data, although not always of very good quality, can be used to reliably assess the impact of fish farming on receiving water quality. The impact of each farm on water quality was assessed, and it was demonstrated that the impact was significant for $NH_4$-N and DO and more important for the farms which produced the most during the investigated period. Those results were expected in regard to other studies performed on the impact of traditional trout production across the world (*Boaventura et al., 1997*; *Caramel et al., 2014*; *Garcia et al., 2014*; *Teodorowicz, 2013*). The cumulative impact of the fish farms on receiving water quality was also assessed and it was demonstrated that the distance between

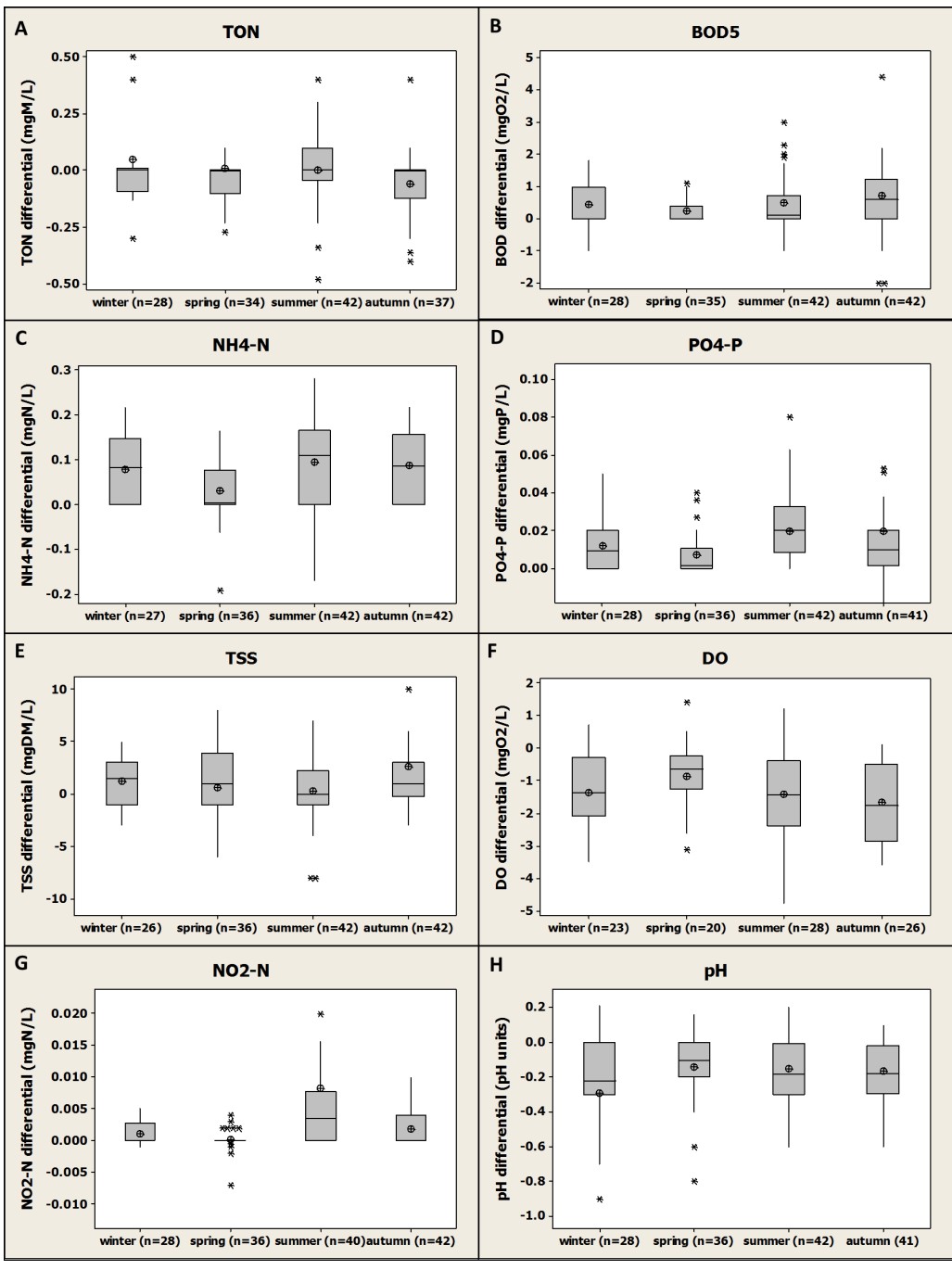

**Figure 5** Box-plot representation of the seasonal variation of the differential values compiled and gathered for the four fish farms for the monitored parameters. Median, average, Q1 and Q3 are presented.

the different fish farms was globally sufficient to allow the river for self-purification regarding the parameters analysed and no overall cumulative impact was observed for the parameters considered. However, considering the relative high water volumes extracted by the studied farm, an increase production would not be possible without the addition of water treatment technologies to apply water reuse and therefore to reduce the demand on water without impacting receiving water quality. The present study represents the first benchmarking of the freshwater fish farming industry in Ireland and will be used as a baseline study, along with a study of the potential impacts on river's hydromorphology, for comparison before the evolution through more advanced practices and the expected implementation of water treatment processes in a near future due to the more and more stringent legislative framework. However, only the relevance of using the water quality parameters can be discussed. Indeed, notions such as the impact on biodiversity and ecotoxicology were not assessed in the present study and will represent a challenge for future studies in order to fully take account of the impact of the aquaculture industry in regard to the implication of the WFD and the expected future intensification of the production. The present study will provide a basis for policy and commensurate decisions on future fish farm licencing and for meeting good quality status under WFD as it pertains to the operation of fish farming facilities.

### Funding
This work was supported by Ireland's Department of Agriculture, Food and the Marine through the Morefish project and by Bord Iascaigh Mhara though the EcoAqua project. The funders had no role in study design, data collection and analysis, decision to publish, or preparation of the manuscript.

### Grant Disclosures
The following grant information was disclosed by the authors:
Ireland's Department of Agriculture, Food and the Marine.
Bord Iascaigh Mhara.

### Competing Interests
The authors declare there are no competing interests.

### Author Contributions
- Alexandre Tahar conceived and designed the experiments, performed the experiments, analyzed the data, contributed reagents/materials/analysis tools, prepared figures and/or tables, authored or reviewed drafts of the paper, approved the final draft.
- Alan M. Kennedy and Richard D. Fitzgerald conceived and designed the experiments, performed the experiments, contributed reagents/materials/analysis tools, authored or reviewed drafts of the paper, approved the final draft.

- Eoghan Clifford and Neil Rowan conceived and designed the experiments, contributed reagents/materials/analysis tools, authored or reviewed drafts of the paper, approved the final draft.

## Data Availability

The raw data are provided as Supplemental Files.

## Supplemental Information

Supplemental information for this article can be found online at http://dx.doi.org/10.7717/peerj.5281#supplemental-information.

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
