# Peer review of "Longitudinal evaluation of the impact of traditional rainbow trout farming on receiving water quality in Ireland"

_PeerJ, doi:10.7717/peerj.5281_

## Round 0.1 · original submission · Major Revisions

Please, read carefully the recommendations given by the reviewers.

As the Academic Editor, I have some additional comments:

Line 196-200: The authors should replace TAN by TON.

Lines 254-257: There are at least three acceptable ways to deal with censored data (two based on simple calculations and one based on statistics). The approach the authors used is not one of them. You assumed concentration as equal to the Limit of Quantification (LOQ) when the analytical method indicated "concentration below LOQ” (< LOQ). I recommend you recalculate these values, informing the approach you selected and the reference.

·

Basic reporting

no comment

Experimental design

no comment

Validity of the findings

no comment

Additional comments

This study represents a 10-year longitudinal survey (2005-2015) of the impact of four consecutive FT trout farms in Ireland based on publically available and independently generated data. The main quality of this study is unique combination of very large time span with data sampled throughout the year, multiple fish farms and relatively complete set of relevant parameters for assessing the water quality. The manuscript is clearly written, data appropriately analyzed and all conclusions strengthen with relevant literature data. Therefore it is my opinion that the manuscript should be accepted for publication in PeerJ after the correction of some minor issues.

The minor corrections are as follows:

General comments:

1.When mentioning average values it should be done in the form value ± SD or SE and if standard error is given than it should be accompanied with the number of samples (n=xx). Please correct that throughout the Results and discussion section.
2. It is customary to write p value as P. Please correct that throughout the Results and discussion section.

Specific comments in the text:

Line 157: there are two full stops, delete one.
Lines 164-173: I think that measured parameters should be listed in line; there is no need for such emphasis.
Lines 197-199: the sentence should be rewritten as follows:
The following range of assessments were considered when evaluating the impacts of the farms on the receiving river water – individual farm impact on river water quality, and cumulative impact of the 4 fish farms on river water quality.
Line 255: add space before the citation
Lines 434-450: please put a statement about statistical significance of changes in NO2-N
Legend for figure 4: change (2015-2015) with (2005-2015)

Reviewer 2 ·

Basic reporting

After careful reading of the manuscript “Longitudinal evaluation of the impact of traditional
rainbow trout farming on receiving water quality in Ireland (#23597)” I have some considerations to make about it.

The study presented in the manuscript can be considered in a correct English. However, does not present deep knowledge about the proposed theme.

The text in general is very extensive and can be much more summarized. In some sections the information are repeated and can be presented and discussed together, due to they reach the same conclusions.

Figures and tables were not well labeled and need more information about them.

I thank you for providing the raw data, however your supplemental files need more information (legends and the values of the LOQ of each parameters) to be useful to future readers.

Experimental design

The research question was defined in the abstract, but not very well defined in the text.

I commend the authors for their extensive data set, compiled over many years. If there is a weakness, it is in the way that the data was analyzed e presented, which should be improved.

The methodology section should also be improved with more information about the farms (number of earthen ponds, volumes, inlet volumes, daily % of renewal, name of the river, among others relevant information).

The showed average inlet data (0.4 m³/s) is not relevant, due to the distinct activities of each farm. I think that this value of 0.4 m³/s is to high for the hatchery and juveniles production.

In the parameters monitored is missing the units of them, and some of them are named and others are in the chemical formulas.

The APHA 2005 is outdated.

The sampling section should be improved in many ways.

In the data treatment, the statistical test should have the description improved (e.g. box plot).

Finally, due to the high volumes of inlet and outlet, I suggest that your data should be analyzed as load instead of concentrations.

Validity of the findings

The presented tables (1-4) are not necessary and the content can be shown in the text.

The figure 1 is missing basic information: extension of the river stretch; inlet volumes of each farm; flow and river velocity; and distance between farms.

The figure 2 should analyze the inlet and outlet of each farm instead of global median.

Figure 3 OK.

Figure 4 should show data of the daily load that is more representative in that high volume,
instead of concentration (e.g. BOD 5: 2 mg O2/ L is the same of 69 kg/day).

I suggest that you analyze and discuss the load instead of concentration, because of the probable high renewal of the earthen ponds. This high renewal of the earthen ponds is diluting the pollutant loads and therefore they were low and not significant. The fish farms use an average of 36.4% (0.4 m³ / L) of the total river flow. When the effluents are released (outlet), they are still diluted in 63.6% (0.7 m³ / s) and they have at least 1km to be consumed by the ecosystem.

I think this is very good point to discuss, the high demand of water of those farms. This eliminates the possibility of future expansions in those fish farming due to the high commitment of water use.

Additional comments

The study presented in the manuscript can be considered relevance. Although, a revision in the way of how the data should be presented, analyzed and discussed is strongly suggested to be more relevant and to have more novelty. The text in general is very extensive and can be much more summarized.

---

## Round 0.2 · accepted · Accept

Once you receive the list of Production tasks, I kindly ask you to check carefully all aspects of the final version to guarantee the highest quality of your article.

#